# A Modified Differential Evolution for Source Localization Using RSS Measurements

**DOI:** 10.3390/s25123787

**Published:** 2025-06-17

**Authors:** Yunjie Tao, Lincan Li, Shengming Chang

**Affiliations:** School of Cyber Science and Engineering, Ningbo University of Technology, Ningbo 315211, China; tyj002915@163.com (Y.T.); csm20130504@163.com (S.C.)

**Keywords:** differential evolution (DE), source localization, received signal strength (RSS), opposition-based learning (OBL), wireless sensor networks (WSNs)

## Abstract

In wireless sensor networks, evolutionary algorithms have emerged as pivotal tools for addressing complex localization challenges inherent in non-convex and nonlinear maximum likelihood estimation problems associated with received signal strength (RSS) measurements. While differential evolution (DE) has demonstrated notable efficacy in optimizing multimodal cost functions, conventional implementations often grapple with suboptimal convergence rates and susceptibility to local optima. To overcome these limitations, this paper proposes a novel enhancement of DE by integrating opposition-based learning (OBL) principles. The proposed method introduces an adaptive scaling factor that dynamically balances global exploration and local exploitation during the evolutionary process, coupled with a penalty-augmented cost function to effectively utilize boundary information while eliminating explicit constraint handling. Comparative evaluations against state-of-the-art techniques—including semidefinite programming, linear least squares, and simulated annealing—reveal significant improvements in both convergence speed and positioning precision. Experimental results under diverse noise conditions and network configurations further validate the robustness and superiority of the proposed approach, particularly in scenarios characterized by high environmental uncertainty or sparse anchor node deployments.

## 1. Introduction

Wireless sensor networks (WSNs), composed of spatially distributed nodes with sensing, computation, and communication capabilities, have become indispensable in applications ranging from environmental monitoring to industrial automation [1]. These networks typically comprise two distinct node categories: anchor nodes (ANs), with precisely known coordinates, and a target node (TN), requiring localization. While WSNs offer substantial benefits in data collection efficiency, their positioning accuracy remains hindered by environmental noise, hardware imperfections, and non-line-of-sight (NLOS) propagation effects [2]. The criticality of accurate localization stems from the spatial dependency of sensor data, where positional context fundamentally determines information relevance in scenarios such as disaster response and military surveillance.

Distance-based localization methodologies, including time of arrival (TOA) [3], time difference of arrival (TDOA) [4], angle of arrival (AOA) [5], and received signal strength (RSS) [6] techniques, have been extensively investigated. Among these, RSS-based approaches dominate practical implementations due to their hardware simplicity and energy efficiency [7]. The RSS localization framework typically formulates a maximum likelihood (ML) estimation problem derived from the log-normal shadowing model [1]. Despite its prevalence, RSS localization has two inherent limitations: sensitivity to path-loss exponent variations and non-convexity in ML estimation when the transmitter power is unknown.

Conventional optimization techniques, such as least squares (LS), weighted least squares (WLS), and semidefinite programming (SDP) [8], often require meticulous initialization to avoid convergence to local minima in multimodal cost landscapes. Evolutionary algorithms—notably differential evolution (DE) [9], particle swarm optimization (PSO) [10], and simulated annealing (SA) [11]—have demonstrated superior global optimization capabilities for such non-convex problems. DE, in particular, has gained prominence due to its balance of computational efficiency and robustness [12]. Recent advancements integrate opposition-based learning (OBL) [13,14,15,16,17] to enhance population diversity, as exemplified in DEOR variants [18]. Nevertheless, two critical limitations persist in existing DE implementations: (1) fixed scaling factors (*F*), that enforce suboptimal exploration–exploitation trade-offs; and (2) restrictive boundary constraints, that disregard valuable information from infeasible solutions during early iterations.

This paper addresses these limitations through two synergistic enhancements to DE. Firstly, an adaptive scaling factor mechanism is proposed. This mechanism adjusts the scaling factor *F* dynamically according to the running state of the algorithm during the optimization process. When F≫0.5, the algorithm focuses on global exploration, widely searching the solution space to obtain the approximate feasible region of the problem. As the optimization progresses, when F≪0.5, the algorithm automatically switches to the local refinement mode, conducting a detailed search for the feasible region obtained earlier to improve the accuracy of the solution. Secondly, a penalty-augmented cost function is introduced. In the early stages of algorithm iteration, feasible solutions close to the boundary are fully utilized to provide directional guidance for the search. At the same time, as the number of iterations increases, this function gradually increases the penalty for constraint violations, effectively guiding the algorithm to find the optimal solution that satisfies the constraint conditions. In addition, in this study, the initial search boundary is strategically broadened to ensure that the algorithm can cover a wider solution space and avoid becoming trapped in local optima. Moreover, the traditional explicit constraint-handling link is removed, simplifying the algorithm process. Through the above improvements, this method can significantly accelerate the convergence speed while ensuring that the quality of the solution is not affected, providing an efficient algorithm support for solving complex optimization problems.

The remainder of this paper is organized as follows: Section 2 formalizes the localization problem and DE fundamentals. Section 3 details the proposed algorithmic enhancements. A parameter sensitivity analysis appears in Section 4, followed by comprehensive performance comparisons in Section 5. Conclusions and future directions are discussed in Section 6.

## 2. Source Localization with Differential Evolution

### 2.1. Localization Problem Formulation

Consider a two-dimensional (2-D) WSNs comprising *N* ANs with known coordinates $x_{n}=\left(x_{n},y_{n}\right)\in R^{2}$ for n=1,…,N, and a TN at unknown position $x_{0}=\left(x_{0},y_{0}\right)$. The RSS at the *n*-th anchor node follows the log-normal shadowing model [1]:(1)$$P_{n}=P_{0}-10\gamma \log_{10}∥x_{n}-x_{0}∥+\chi_{n},$$
where $∥\cdot ∥$ denotes the Euclidean norm, $P_{0}$ denotes the transmit power, γ is the path-loss exponent, and $\chi_{n}∼N\left(0,\sigma_{n}^{2}\right)$ represents log-normal shadowing noise. The localization task reduces to solving the ML estimation problem:(2)$${\hat{x}}_{0}=\underset{x}{arg min}f\left(x\right),$$
with the ML cost function defined as(3)$$f\left(x\right)=\sum_{n=1}^{N}\frac{{\left(P_{n}-P_{0}+10\gamma \log_{10}∥x_{n}-x∥\right)}^{2}}{\sigma_{n}^{2}},$$
where ${\hat{x}}_{0}$ is the estimated position, and both *x* and $x_{n}$ are vectors in $R^{2}$.

### 2.2. Differential Evolution Framework

DE is a population-based optimization technique that works through successive generations of candidate solutions. The algorithm begins by creating an initial random population within the specified search boundaries. It then iteratively improves the solutions through its unique combination of evolutionary operations, carefully balancing global exploration with local refinement. A key feature is its selection mechanism that consistently preserves better solutions while discarding inferior ones. This approach allows the method to efficiently search complex spaces while maintaining diversity and avoiding early convergence to suboptimal solutions.

Firstly, we use a set $I_{n}^{g}$ to represent the *n*-th individual in generation *g*, which represents a 2-D vector $\left(I_{n1}^{g},I_{n2}^{g}\right)$, representing the candidate for the estimated position of the TN in the localization problem. These initial individuals must constantly move towards the unknown position $x_{0}$ in the solution space through the three processes of mutation, crossover, and selection. The solution space can be expressed in the following form:(4)$$Z=\left\{\left(z_{1},z_{2}\right)|a_{1}<z_{1}<b_{1},a_{1}<z_{2}<b_{1}\right\},$$

In the optimization framework, $a_{1}$ and $b_{1}$ represent the upper and lower bounds of the solution space, respectively, defining the constrained domain for all feasible solutions. We formally denote the initial population as $I_{1}^{1}$, which corresponds to the first generation of individuals in the evolutionary process. This notation specifically refers to the population that needs to be properly initialized before the algorithm begins its search procedure. The initialization of $I_{1}^{1}$ involves the generation of candidate solutions within the prescribed limits $\left[a_{1},b_{1}\right]$, establishing the starting point for subsequent optimization operations.

#### 2.2.1. Mutation Operator

${\hat{I}}_{n}^{g}$ is generated from the initial population using the following mutation methods:(5)$${\hat{I}}_{n}^{g}=I_{k}^{g}+F\left(I_{p}^{g}-I_{q}^{g}\right),$$
where *F* denotes the scaling factor in the DE algorithm, which plays a crucial role in controlling the mutation operation. The value of *F* directly determines the magnitude of the variation range during the mutation process. When the scaling factor *F* assumes a relatively small value (typically F<0.5), the corresponding variation range becomes constrained. In such cases, the algorithm exhibits stronger local search characteristics, demonstrating a pronounced tendency to explore and exploit the neighborhood of current solutions within a limited search space. This behavior enhances the algorithm’s ability to perform intensive local search and fine-tune solutions. Conversely, when *F* takes a larger value (generally F>0.8), it results in an expanded mutation range. Under these circumstances, the algorithm manifests more prominent global search properties, showing greater propensity to explore distant regions in the solution space. This characteristic improves the algorithm’s capacity to escape local optima and discover potentially better solutions in broader search areas. It is worth noting that the selection of an appropriate *F* value requires careful consideration, as it represents a trade-off between the algorithm’s local exploitation capability and global exploration potential. The optimal setting of *F* may vary depending on specific problem characteristics and should be determined through empirical testing or adaptive mechanisms.

#### 2.2.2. Crossover Operation

Next, we cross the individual obtained by mutation and the original individual to obtain a new individual $U_{n}^{g}$:(6)$$U_{n}^{g}=\left\{\begin{array}{cc} {\hat{I}}_{n}^{g}, & rand\leq CR\\ I_{n}^{g}, & rand>CR \end{array}\right.,$$
where rand represents a uniformly distributed random variable sampled from the interval [0,1], which is independently generated during each iteration of the algorithm. CR denotes the crossover rate, a critical control parameter that governs the genetic recombination process by probabilistically determining whether to preserve the genetic material of the parent individual or to incorporate the genetic material of the mutant individual. This fundamental mechanism enables DE to effectively balance exploration and exploitation during the optimization process.

#### 2.2.3. Selection Mechanism

The new individual after crossover is compared to the original individual and the fitness value of each individual is calculated using the fitness function. The latest individual with the smallest fitness value is selected:(7)$$I_{n}^{g+1}=\arg\min_{I_{n}^{g},U_{n}^{g}}\left\{f\left(I_{n}^{g}\right),f\left(U_{n}^{g}\right)\right\}.$$

After repeating for *G* generations, the retained individual $I_{DE}$ is the estimated position of the TN:(8)$$I_{DE}=\arg\min_{I_{n}^{G}}\left\{f\left(I_{n}^{G}\right)\right\}.$$

## 3. Enhanced Differential Evolution Framework

To address the limitations of conventional DE in RSS-based localization, we propose two key algorithmic improvements that significantly accelerate the convergence speed while improving the positioning accuracy. In this paper, the improved differential evolution is called MDE. First, based on OBL principles, we develop an adaptive scaling factor mechanism that dynamically adjusts the conventional scaling factor during the optimization process. Second, we introduce an improved cost function formulation by incorporating a penalty term into the original objective function, which enables a more accurate fitness evaluation during evolutionary computation. These innovations collectively improve global search capability while maintaining solution precision.

### 3.1. Opposition-Based Population Initialization

We randomly generate *n* original individuals within a given search range, which is recorded as the first-generation population $I_{n}^{1}$. It is worth noting that in this paper we appropriately expand the given search range. We refer to the scaling factor for boundary expansion as α, which is explained in detail later. Then, another generation of population ${\tilde{I}}_{n}^{1}$ is obtained using the OBL processing method. These two population generations are centrosymmetric in search range. Then, the two population generations are merged, $V_{n}^{1}=\left(I_{n}^{1},{\tilde{I}}_{n}^{1}\right)$, and the fitness value of each individual is calculated by the cost function (Equation 11), and then sorted from small to large, and the first *n* individuals ${\tilde{V}}_{n}^{1}$ are retained.

### 3.2. Adaptive Mutation Operator

According to the mutation formula,(9)$${\ddot{V}}_{n}^{1}={\tilde{V}}_{n}^{1}\left(k\right)+F\left({\tilde{V}}_{n}^{1}\left(p\right)-{\tilde{V}}_{n}^{1}\left(q\right)\right),$$
a new population set is obtained, where *F* is an adaptive scaling factor:(10)$$F=F_{0}2^{e^{\frac{1-G}{1+G-g}}},$$
where $F_{0}$ represents the initial scaling factor value, *G* denotes the maximum number of generations (total population count), and *g* indicates the current generation index in the evolutionary process. As clearly demonstrated by the mathematical relationship, during the early evolutionary stages (g≪G), the scaling factor *F* maintains relatively higher values, consequently producing larger mutation step sizes. This characteristic facilitates an extensive exploration of the solution space. As evolution progresses to later stages (g→G), *F* gradually decreases, resulting in finer, more localized mutation steps. This behavior enhances the algorithm’s ability to perform intensive local search around promising regions. This adaptive mechanism provides two key optimization benefits: First, a significant improvement in the initial population diversity through global exploration. Second, a precise refinement of the local exploitation capability during the final convergence. Compared to traditional DE, the adaptive scaling factor can expand the mutation step size in the early stages to quickly cover potential optimal regions, while refining the search in later stages, effectively accelerating convergence speed. Traditional DE requires manual tuning of the scaling factor, preventing the method from adapting well to dynamic environments.

### 3.3. Crossover

The original population and the mutated population are crossed using the traditional DE method, which is the same as the method in Section 2. The crossed population is recorded as *U*.

### 3.4. Selection

The fitness value of the crossed population is calculated by a new cost function,(11)$$\hat{f}\left(x\right)=\sum_{n=1}^{N}\frac{1}{{\sigma_{n}}^{2}}{\left(P_{n}-P_{0}+10\gamma \log_{10}∥x_{n}-x∥\right)}^{2}+M\dot{f},$$
where *M* is the penalty factor,(12)$$M={\frac{g}{2}}^{2},$$
and *M* increases with the increase in the number of iterations. The penalty term represents the degree of individual violation of the constraint, because when the population is initialized, we appropriately expand the search range, so there will be some individuals outside the search range. For these individuals,(13)$$\dot{f}=∥U_{n}^{g}-\left(\frac{a_{1}+b_{1}}{2},\frac{a_{1}+b_{1}}{2}\right)∥.$$

However, for those individuals who are within the constraints themselves,(14)$$\dot{f}=0.$$

Finally, the minimum fitness value is selected as the new initial point, and the above steps are repeated until the number of iterations reaches the algebra *G* of the population. The individual with the smallest fitness value in the last generation is used as the estimated position of the TN:(15)$$I_{MDE}=\arg\min_{U_{n}^{G}}\left\{\hat{f}\left(U_{n}^{G}\right)\right\}.$$

Figure 1 illustrates the architectural workflow of the proposed enhanced differential evolution algorithm, with its corresponding pseudo-code formalization provided in Algorithm 1.
**Algorithm 1** Pseudo-code of the Proposed MDE Algorithm**Input:** Locations $x_{i}$ and number *N* of anchor nodes, crossover probability CR, the number of iterations of the population *G*, scaling factor $F_{0}$, extension factor $a_{0}$, population size *n***Output:** 
Estimated location $\hat{x}$ of the target node1:Initialization//generate initial population2:Get a new generation of populations ${\tilde{V}}_{n}^{1}$//Two vectors $V_{n}^{1}=\;\left[I_{n}^{1},{\tilde{I}}_{n}^{1}\right]\;$ are merged after using OBL, and the first *n* vectors are selected by the cost function (Equation 4).3:Repeat *G* times//Mutate via (Equation 9)//Perform crossover as (Equation 6)//Select best individual with (Equation 11)4:Select $I_{MDE}$ with (Equation 15)//generate mutated population5:End

## 4. Parameter Sensitivity Analysis

It is well known that the DE algorithm involves multiple key parameters that significantly influence its performance and convergence behavior. Among these variables, the number of iterations of the population *G* plays a crucial role in determining the maximum number of evolutionary generations that the algorithm undergoes. Furthermore, the population size NP, which represents the number of individuals in each generation, directly affects the diversity and exploration capacity of the algorithm. Another essential parameter is the probability of crossover CR, which controls the rate at which genetic information is exchanged between candidate solutions during the recombination phase. Furthermore, in our proposed enhanced DE algorithm, we introduce an adaptive scaling factor $F_{0}$ as its initial value, which dynamically adjusts to improve the efficiency of convergence. Alongside this, we incorporate an expansion factor α, which is utilized to modify the constraint conditions, thereby enhancing the algorithm’s flexibility and robustness in handling optimization problems under varying conditions. In this section, we study the values of each variable in turn and find the values that improve the performance of DE. Through systematic experimentation, we investigate the effects of four critical parameters. All experiments maintain constant environmental conditions (σ = 2, γ = 3, N = 10) within a $40\times 40\;m^{2}$ search domain. The performance criterion is the root mean square error (RMSE), which is defined as(16)$$\mathrm{RMSE}=\sqrt{\frac{1}{M_{c}}\sum_{i=1}^{M_{c}}{∥x_{i}-{\hat{x}}_{i}∥}^{2}}.$$

All comparative experiments adopt the optimized parameters from Section 4 unless otherwise specified. The environmental parameters follow the log-normal shadowing model (Equation 1) and real-world restaurant deployment constraints. For ease of reference, all the parameters are presented in Table 1.

### 4.1. Population Size Optimization

Figure 2 demonstrates the non-monotonic relationship between population size (NP) and localization accuracy RMSE. In this figure, we first take CR=0.9, according to the empirical value, and then randomly select the values of *G*, $F_{0}$, and α. Extensive experimental results demonstrate that the population size parameter in the DE algorithm exhibits a non-trivial impact on optimization performance. Specifically, our systematic investigations reveal that selecting excessively small or unduly large values for the individual count (NP) in the population leads to suboptimal algorithmic performance. When the population contains too few individuals (that is, NP is set below a certain threshold), the algorithm fails to maintain sufficient genetic diversity, limiting its exploration ability in the solution space. In contrast, a population size that is too large (that is, NP exceeds an optimal range) results in unnecessary computational overhead without providing the corresponding improvements in solution quality. Through comprehensive testing across multiple benchmark functions, we have quantitatively verified that there exists an optimal range for the individual count that maximizes the performance enhancement potential of the DE algorithm. Deviating from this optimal range in either direction, whether by choosing a value that is too conservative or too aggressive, will inevitably compromise the algorithm’s ability to leverage its full performance improvement capabilities. When NP=200, performance is the best in our experiment. Then, we determined the value of NP, and other variables remained fixed. The experiments were repeated to study the influence of the number of iterations of the population *G* on DE.

### 4.2. Generation Limit Selection

Figure 3 shows that the number of iterations of the population also affects DE to a greater or lesser extent, and the experimental results are the best when G=30 in our experiment. Next, we determine *n* and *G*, and we discuss whether the value of $F_{0}$ also affects the experiment.

### 4.3. Scaling Factor Adaptation

The non-convex RMSE landscape in Figure 4 reveals an optimal initial scaling factor $F_{0}=0.5$. This value enables a smooth transition from global exploration (F>0.8 in early generations) to local refinement (F<0.5 in final stages), as governed by the adaptive law in (Equation 10).

### 4.4. Boundary Expansion Analysis

Figure 5 demonstrates the critical role of the expansion factor α in balancing the exploration capability and computational efficiency. The experimental results reveal an optimal boundary expansion ratio at α=1.4. This configuration achieves a minimum RMSE of 0.5 m by effectively utilizing potential solutions near constraint edges while avoiding excessive computational overhead from over-expanded search domains.

Through a systematic parameter sensitivity analysis, we establish the empirically determined optimal parameters: NP=200, G=30, $F_{0}=0.5$, α=1.4. These are used to compare the performance of MDE with other algorithms.

## 5. Performance Evaluation and Comparative Analysis

### 5.1. Experimental Validation

The proposed MDE algorithm is rigorously evaluated in realistic indoor positioning scenarios, specifically addressing the operational challenges of intelligent food delivery robots in commercial restaurants.

In practical restaurant deployment scenarios, intelligent food delivery robots face significant positioning challenges when performing order delivery tasks. When receiving an order, the robot must first achieve real-time self-localization to determine its current position within the restaurant’s dynamic environment, while simultaneously identifying the customer’s location through a multisensor fusion system. The positioning infrastructure comprises a distributed network of strategically placed wireless anchors and beacons on the ceiling at regular intervals, forming a sensor array that ensures full coverage of the area. However, the random initial position of the robot and the constantly changing layout of the restaurant, with moving furniture, staff, and other dynamic obstacles, create complex signal propagation conditions where the number and spatial distribution of detectable sensors vary significantly. These challenges become particularly acute in large dining establishments during peak operating hours, where high customer density leads to signal interference, frequent blockages, and variable noise levels in different frequency bands. Under such demanding conditions, the robot’s positioning performance in terms of both accuracy and speed is crucial for its operational effectiveness. The system’s ability to determine its position effectively directly impacts its path planning efficiency, obstacle avoidance reliability, and ultimately the success rate of food delivery. This makes the development of robust localization algorithms essential for ensuring operation in commercial environments, where system performance affects service quality. The proposed enhanced positioning system MDE algorithm addresses these operational requirements through its adaptive processing capabilities and optimized computational efficiency.

To compare the positioning performance of MDE, we compare the LSRE method based on error estimation (WLS) [8], SDP [7], and localization based on simulated annealing optimization (SA) [11], standard DEOR [18], and improved DEOR-F [19]. For the sake of convenience, we list all the methods that are compared in Table 2. The algorithms of these methods were implemented in MATLAB R2021a. For the SDP method, we used the well-known CVX software package [20].

To generate RSS measurements, we set the path loss $P_{0}=-10$ dBm, the path-loss exponent γ=3, and the reference distance $d_{0}=1$ m. In all the experiments in this paper, the number of Monte Carlo runs is $M_{c}=10,000$. The anchor nodes and target node are randomly distributed in a square area of $40\times 40\;m^{2}$. And the target node can communicate with all the anchor nodes. The performance criterion is the RMSE.

Figure 6 shows a comparison between the performance of our proposed method and several other methods when the number of AN nodes is fixed, with increasing noise. The results show that our proposed method is superior to the traditional methods, having a smaller RMSE. Our proposed algorithm demonstrates superior performance compared to other population-based approaches (DEOR, SA, and DEOR-F) in low-noise environments.

In Figure 7, we show that when the noise is constant, the RMSE changes with an increase in the number of anchor nodes ANs. The MDE algorithm yields lower RMSE values with increasing numbers of anchor nodes. It should be noted that these suggested parameter values (namely, of α, CR, NP, and *G*) are not absolute or universally valid, as their appropriate settings fundamentally depend on the characteristics of each particular problem.

### 5.2. Comparative Analysis

The proposed MDE algorithm demonstrates superior performance compared to traditional positioning methods due to its inherent population-based heuristic characteristics and strong global optimization capabilities, making it particularly effective for addressing complex positioning problems. Our improved DE strategy achieves a faster convergence speed than conventional DE implementations through several key enhancements. The incorporation of an adaptive scaling factor *F* significantly improves the algorithm’s flexibility by dynamically adjusting the mutation step sizes during the evolutionary process, enabling more efficient search range adaptation that accelerates convergence toward the global optimum. Furthermore, we introduce a novel cost function that serves a dual purpose: it not only eliminates the need for separate boundary processing but also actively utilizes boundary information to guide the global search toward optimal solutions.

Regarding computational complexity, the computational complexity analyses of all the algorithms are summarized in Table 3. Specifically, SDP exhibits the highest complexity with substantial computational resource consumption. While WLS demonstrates the lowest complexity, its positioning accuracy is insufficient. Similarly, SA also features relatively low complexity, though its stochastic search nature may imply a larger constant factor. The proposed method shows comparable complexity to both DEOR and DEOR-F, with all three methods primarily depending on the number of iterations (*G*), population size (NP), and anchor node count (*N*). This clearly indicates that the introduced adaptive mechanism does not significantly increase computational overhead. The synergistic combination of these enhancements results in a more robust and efficient positioning solution that effectively balances exploration and exploitation throughout the optimization process.

Finally, in Table 4, we present the average computation times of all the algorithms. While our proposed method, which incorporates OBL and employs an adaptive scaling factor, along with the innovative introduction of a novel penalty function in the cost function, results in slightly higher computation times compared to DEOR and DEOR-F, its superior localization accuracy sufficiently compensates for this marginal increase in runtime.

### 5.3. Convergence Analysis

Table 5 presents the convergence performance of MDE in comparison with DEOR and DEOR-F under the fixed conditions of N=10 anchor nodes and a population size of NP=80. All the algorithms were executed under identical settings to ensure fair benchmarking. From the results, it is observed that DEOR-F achieves the fastest convergence (8 generations), followed by DEOR (10 generations) and MDE (12 generations). However, despite requiring slightly more iterations to converge, MDE yields the lowest RMSE, demonstrating better positioning accuracy. This trade-off in convergence speed is primarily attributed to the adaptive scaling factor employed by MDE. In the early stages of the evolutionary process, MDE maintains a relatively large scaling factor *F*, promoting broad exploration of the search space and enhancing global search capabilities. As iterations progress, *F* gradually decreases, shifting the search behavior toward local refinement. This dynamic adjustment helps avoid premature convergence but slightly delays reaching the termination criterion.

It is worth noting that the added complexity introduced by the adaptive mechanism does not increase computational overhead. As shown in the runtime column, the execution time of MDE remains comparable to DEOR and DEOR-F, validating the efficiency of the proposed strategy.

## 6. Conclusions

This paper presents a systematic advancement in RSS-based localization through a modified differential evolution framework that addresses two fundamental challenges in evolutionary optimization: the exploration–exploitation dilemma and constraint boundary utilization. By integrating an adaptive scaling factor with opposition-based learning principles, the proposed algorithm dynamically adjusts its search strategy during the optimization process, enabling robust performance across diverse operational environments. The novel penalty-augmented cost function further enhances positioning accuracy by strategically incorporating spatial constraints without imposing restrictive boundary conditions.

Experimental validation confirms significant improvements in both accuracy and efficiency: (1) Superior robustness under challenging noise conditions compared to state-of-the-art evolutionary methods (DEOR, DEOR-F), particularly where traditional approaches deteriorate; (2) accelerated convergence relative to convex optimization and stochastic search techniques (SDP, SA), achieving real-time capability critical for dynamic applications; (3) computational efficiency on par with advanced DE variants despite enhanced mechanisms, maintaining scalability for large-scale networks. The numerical results substantiate these improvements: Under high noise (σ=3 dB), MDE achieves an RMSE of 2.2 m, outperforming DEOR-F (2.3 m) and SA (2.4 m). With N=10 anchors, MDE reduces the RMSE to 2.8 m, exceeding SDP (3.5 m) and WLS (4.0 m). Computationally, MDE maintains O(G·NP·N) complexity with an average runtime of 0.0048 s—100× faster than SDP ($O(N^{3.5})$). Although requiring 12 generations for convergence (vs. DEOR-F’s 8), MDE achieves the lowest RMSE (2.96 m) under identical settings.

The synergistic combination of population diversity preservation and adaptive parameter control enables efficient navigation of multimodal solution spaces. Looking forward, this work establishes a foundation for several promising research directions. Extending the framework to three-dimensional environments could address multi-floor localization challenges in smart infrastructure applications. The integration of real-time parameter adaptation mechanisms may further improve mobile target tracking capabilities in dynamic scenarios. Additionally, exploring hybrid architectures that combine evolutionary optimization with deep learning techniques could unlock new possibilities for autonomous navigation systems. The methodology’s inherent flexibility suggests broader applicability beyond wireless sensor networks, potentially benefiting emerging domains such as swarm robotics and industrial IoT deployments. While this study focuses on algorithmic innovation under standardized models, future work will incorporate public RSS datasets to quantify NLOS mitigation gains.

## Figures and Tables

**Figure 1 sensors-25-03787-f001:**
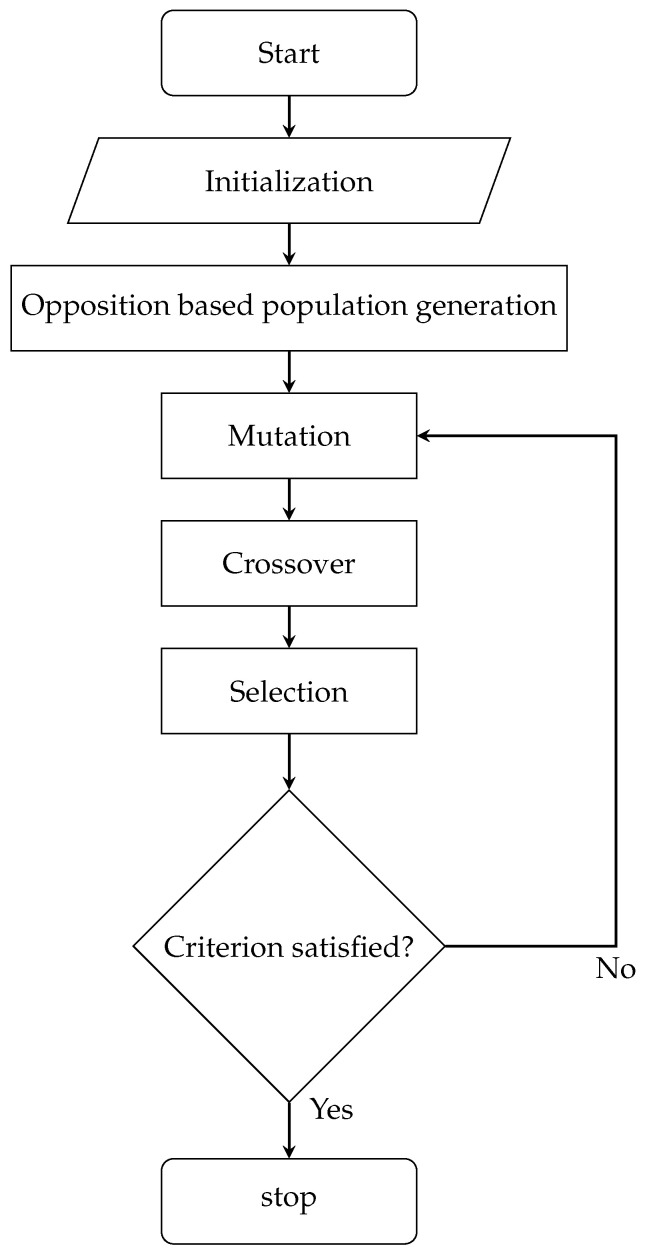
Flowchart of the MDE.

**Figure 2 sensors-25-03787-f002:**
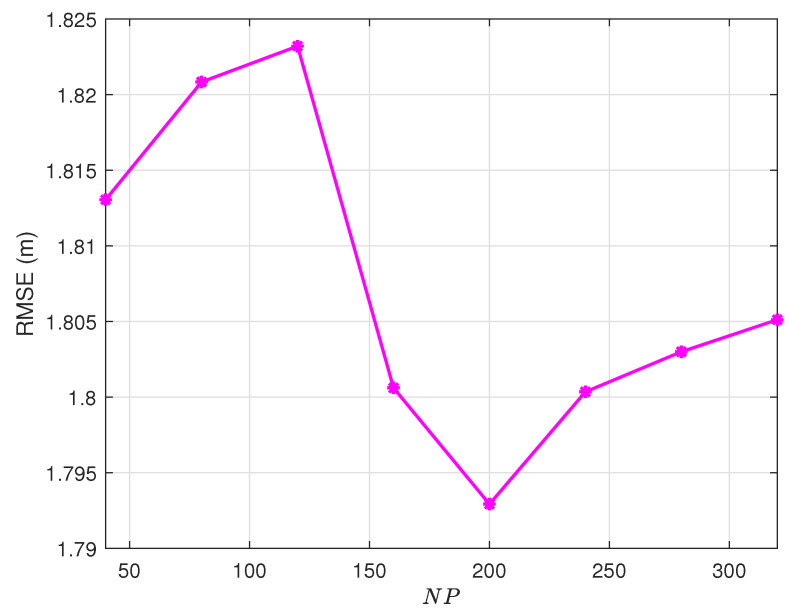
Population size sensitivity: RMSE vs. NP (G = 50, $F_{0}\;=\;0.9$, α = 2, CR = 0.9).

**Figure 3 sensors-25-03787-f003:**
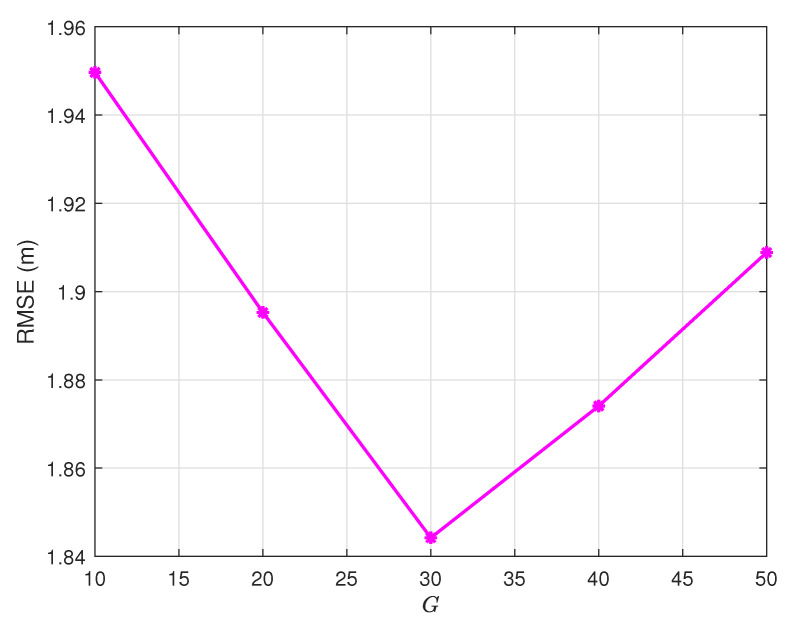
Convergence analysis: RMSE vs. maximum generations *G* (NP=200, $F_{0}=0.9$, α=2, CR=0.9).

**Figure 4 sensors-25-03787-f004:**
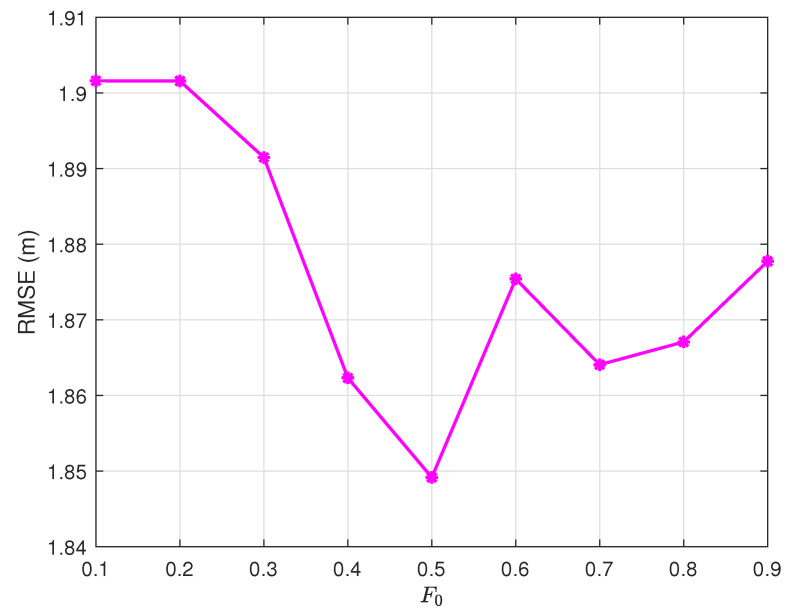
Scaling factor optimization: RMSE vs. $F_{0}$ (NP=200, G=30, α=2, CR=0.9).

**Figure 5 sensors-25-03787-f005:**
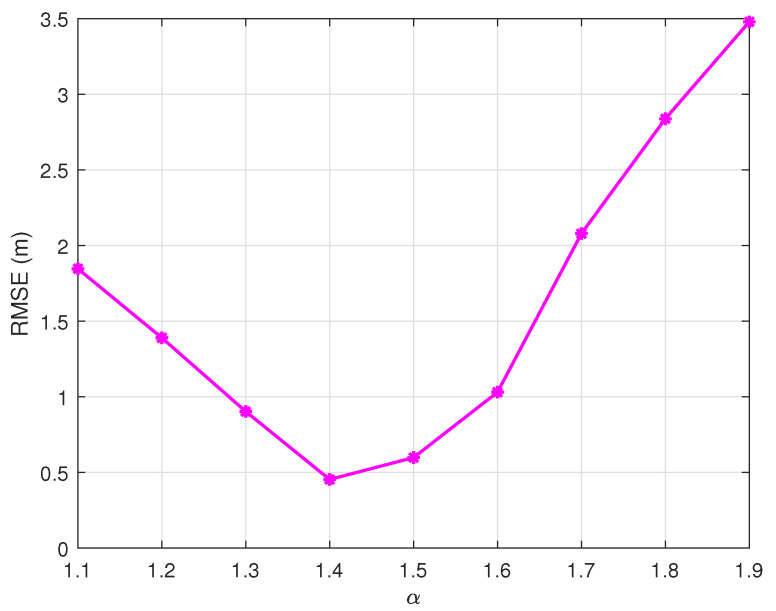
Boundary expansion trade-off: RMSE vs. α (NP=200, G=30, $F_{0}=0.5$, CR=0.9).

**Figure 6 sensors-25-03787-f006:**
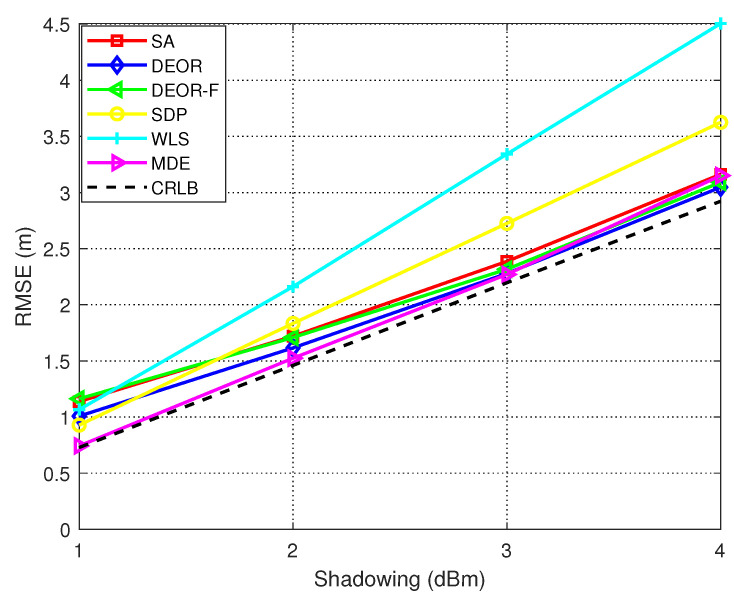
RMSE versus variance of log-shadowing when N=10, G=30, CR=0.9, $F_{0}=0.5$, α=1.4, NP=200.

**Figure 7 sensors-25-03787-f007:**
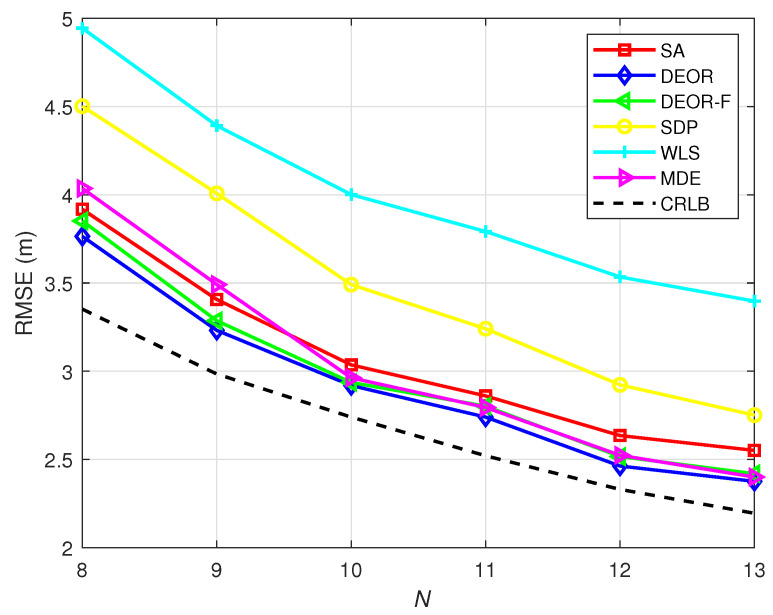
RMSE versus the number of ANs when G=30, CR=0.9, $F_{0}=0.5$, α=1.4, NP=200, σ=2.

**Table 1 sensors-25-03787-t001:** Summary of experimental parameters.

Parameter Name	Symbol	Value/Range	Remarks
Population size	NP	200	Sensitivity analysis (Section 4.1)
Maximum generations	*G*	30	Convergence analysis (Section 4.2)
Initial scaling factor	$F_{0}$	0.5	Parametric optimization (Section 4.3)
Boundary expansion factor	α	1.4	Boundary trade-off (Section 4.4)
Crossover probability	CR	0.9	Empirically determined value
Path-loss exponent	γ	3	Fixed environmental parameter
Reference distance	$d_{0}$	1 m	Fixed environmental parameter
Transmit power	$P_{0}$	−10 dBm	Fixed environmental parameter
Noise standard deviation	σ	1–6 dB	Fixed (varied in some experiments)
Monte Carlo runs	$M_{c}$	10,000	Number of repeated experiments
Search area	–	$40\times 40\;m^{2}$	2-D square region
Anchor node count	*N*	8–13	Fixed/variable parameter

**Table 2 sensors-25-03787-t002:** Comparison of localization methods: Benchmarking against MDE.

Method	Description
MDE	The proposed enhanced differential evolution method for localization.
WLS	Least squares based on error estimation, from [8], considering error factors for localization.
SDP	Semidefinite programming method in [7], handling constraints and objectives for localization.
SA	Localization via simulated annealing optimization, as in [11], for searching optimal solutions.
DEOR	Standard differential evolution with opposition and redirection, from [18], using relevant strategies for localization.
DEOR-F	Improved DEOR method in [19], with extra features for better performance.

**Table 3 sensors-25-03787-t003:** Comparison of computational complexity.

Method	Complexity
MDE	O(G·NP·N)
WLS [8]	O(N)
SDP [7]	$O(N^{3.5})$
SA [11]	O(N)
DEOR [18]	O(G·NP·N)
DEOR-F [19]	O(G·NP·N)

**Table 4 sensors-25-03787-t004:** Average execution time of the algorithms.

Method	MDE	WLS	SDP	SA	DEOR	DEOR-F
Time (s)	0.0048	0.0002	0.3160	0.0011	0.0029	0.0017

**Table 5 sensors-25-03787-t005:** Performance benchmarking against OBL-DE variants.

Algorithm	Convergence Generation	RMSE (m)	Runtime (s)
MDE	12	2.9596	0.0048
DEOR	10	3.1395	0.0029
DEOR-F	8	4.1858	0.0017

## Data Availability

No new data were created or analyzed in this study. Data sharing is not applicable to this article.

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
