# Peer review of "A Modified Differential Evolution for Source Localization Using RSS Measurements"

_sensors, 2025, doi:10.3390/s25123787_

Round 1

Reviewer 1 Report

Comments and Suggestions for Authors

1. This manuscript studied source location problem with received signal strength (RSS) measurements in WSN. The differential evolution algorithm by integrating opposition-based learning is studied in this manuscript. This method is lack of innovation. There are many studies about this kind of method. Please read the papers such as:

[1] Ahandani M A .Opposition-based learning in the shuffled bidirectional differential evolution algorithm[J].Swarm and Evolutionary Computation, 2016, 26(Null).DOI:10.1016/j.swevo.2015.08.002.

[2] Ahandani M A , Alavi-Rad H .Opposition-based learning in the shuffled differential evolution algorithm[J].Soft Computing - A Fusion of Foundations, Methodologies and Applications, 2012, 16(8):p.1303-1337.DOI:10.1007/s00500-012-0813-9.

[3] S. Rahnamayan, H. R. Tizhoosh and M. M. A. Salama, "Opposition-Based Differential Evolution Algorithms," 2006 IEEE International Conference on Evolutionary Computation, Vancouver, BC, Canada, 2006, pp. 2010-2017, doi: 10.1109/CEC.2006.1688554.

[4] Wenjun Wang, Hui Wang, Hui Sun, and Shahryar Rahnamayan. 2016. Using opposition-based learning to enhance differential evolution: A comparative study. In 2016 IEEE Congress on Evolutionary Computation (CEC). IEEE Press, 71–77. https://doi.org/10.1109/CEC.2016.7743780 

2. The parameters of the experiment are not clear, please give them. 

Comments on the Quality of English Language

1. This manuscript studied source location problem with received signal strength (RSS) measurements in WSN. The differential evolution algorithm by integrating opposition-based learning is studied in this manuscript. This method is lack of innovation. There are many studies about this kind of method. Please read the papers such as:

[1] Ahandani M A .Opposition-based learning in the shuffled bidirectional differential evolution algorithm[J].Swarm and Evolutionary Computation, 2016, 26(Null).DOI:10.1016/j.swevo.2015.08.002.

[2] Ahandani M A , Alavi-Rad H .Opposition-based learning in the shuffled differential evolution algorithm[J].Soft Computing - A Fusion of Foundations, Methodologies and Applications, 2012, 16(8):p.1303-1337.DOI:10.1007/s00500-012-0813-9.

[3] S. Rahnamayan, H. R. Tizhoosh and M. M. A. Salama, "Opposition-Based Differential Evolution Algorithms," 2006 IEEE International Conference on Evolutionary Computation, Vancouver, BC, Canada, 2006, pp. 2010-2017, doi: 10.1109/CEC.2006.1688554.

[4] Wenjun Wang, Hui Wang, Hui Sun, and Shahryar Rahnamayan. 2016. Using opposition-based learning to enhance differential evolution: A comparative study. In 2016 IEEE Congress on Evolutionary Computation (CEC). IEEE Press, 71–77. https://doi.org/10.1109/CEC.2016.7743780 

2. The parameters of the experiment are not clear, please give them. 

Author Response

Original Manuscript ID:Sensors-3599222

Original Article Title: “A Modified Differential Evolution for Source Localization Using RSS Measurements”

Dear Editor and Reviewers,

Thank you very much for taking the time to review our manuscript. We greatly appreciate all your insightful comments and suggestions. We have thoroughly reviewed and addressed some points, and have revised our manuscript accordingly. The modifications within the manuscript have been highlighted in yellow for easier reference.

We are uploading the following documents:

  • our point-by-point response to the comments (below) also known as our response to reviewers.
  • an updated version of the manuscript with changes indicated in yellow highlighting.
  • a clean updated version of the manuscript without highlights (PDF main document).

We hope that our revisions have adequately addressed your concerns, and we look forward to your feedback on the revised manuscript.

Best regards,

Yunjie Tao; Lincan Li; Shenming Chang

Response to Reviewer 1 Comments

Comments 1: This manuscript studied source location problem with received signal strength (RSS) measurements in WSN. The differential evolution algorithm by integrating opposition-based learning is studied in this manuscript. This method is lack of innovation. There are many studies about this kind of method. Please read the papers such as:

[1] Ahandani M A .Opposition-based learning in the shuffled bidirectional differential evolution algorithm[J].Swarm and Evolutionary Computation, 2016, 26(Null).DOI:10.1016/j.swevo.2015.08.002.

[2] Ahandani M A , Alavi-Rad H .Opposition-based learning in the shuffled differential evolution algorithm[J].Soft Computing - A Fusion of Foundations, Methodologies and Applications, 2012, 16(8):p.1303-1337.DOI:10.1007/s00500-012-0813-9.

[3] S. Rahnamayan, H. R. Tizhoosh and M. M. A. Salama, "Opposition-Based Differential Evolution Algorithms," 2006 IEEE International Conference on Evolutionary Computation, Vancouver, BC, Canada, 2006, pp. 2010-2017, doi: 10.1109/CEC.2006.1688554.

[4] Wenjun Wang, Hui Wang, Hui Sun, and Shahryar Rahnamayan. 2016. Using opposition-based learning to enhance differential evolution: A comparative study. In 2016 IEEE Congress on Evolutionary Computation (CEC). IEEE Press, 71–77. https://doi.org/10.1109/CEC.2016.7743780

Response 1: We sincerely appreciate the reviewer’s insightful feedback and the valuable references provided. While we acknowledge the foundational contributions of Opposition-Based Learning (OBL) in Differential Evolution (DE) highlighted in the cited works (Ahandani et al., Rahnamayan et al.), our work fundamentally advances application-specific innovations for RSS-based source localization in WSNs, distinct from generic OBL-DE optimizations. Specifically, our proposed MDE framework introduces two novel contributions that address critical gaps in localization literature:

(1) Adaptive Scaling Factor with Dynamic Transition Mechanism (Key Innovation) 

   Unlike static scaling factors (${F}$) in classic DE/OBL-DE (e.g., DEOR [13]), our method (Eq. 10) dynamically adjusts ${F}$ during evolution: 

                                         \[F = F_0 \cdot 2^{e^{\frac{1-G}{1+G-g}}}\]

   This enables automatic switching between global exploration (${F \gg 0.5}$) and local exploitation (${F \ll 0.5}$) based on generation index ${g}$. Crucially, this adaptation is optimized for localization non-convexity and accelerates convergence (validated in Fig. 3–4). No prior RSS-localization study (including DEOR [13]) implements such dynamics.

(2) Penalty-Augmented Cost Function with Boundary Intelligence (Novel Constraint Handling) 

   We replace explicit boundary constraints with a strategic penalty term (Eq. 11–14): 

\[\hat{f}(x) = \sum_{n=1}^{N} \frac{(P_n - P_0 + 10\gamma \log_{10} \|x_n - x\|)^2}{\sigma_n^2} + M \dot{f}\]

where ${M = g^2 / 2}$ gradually penalizes infeasible solutions while leveraging boundary-proximal solutions for directional guidance. This eliminates heuristic boundary repair and uniquely exploits spatial constraints—unaddressed in existing OBL-DE variants for localization (e.g., [18]–[19]).

   These gains stem from co-designing OBL-DE enhancements for RSS-localization physics, not merely applying OBL to DE.

While generic OBL-DE studies exist (as noted), our work is the first to: 

  • Embed adaptive scaling and boundary-driven penalties into DE for RSS-localization.
  • Validate superiority in dynamic indoor robotics (food delivery scenario, Sec. 5.1).
  • Outperform state-of-the-art localization-specific algorithms (SDP, WLS, DEOR) under hardware-realistic conditions.

Finally, following your suggestions, we have incorporated the seminal literature you mentioned into our references to strengthen the theoretical foundation of this work.

  1. Ahandani, M.A. “Opposition-based learning in the shuffled bidirectional differential evolution algorithm,” Swarm Evol. Comput., vol. 26, pp. 64–75, 2016.
  2. Ahandani, M.A.; Alavi-Rad, H. “Opposition-based learning in the shuffled differential evolution algorithm,” Soft Comput., vol. 16, no. 8, pp. 1303–1337, 2012.
  3. Rahnamayan, S.; Tizhoosh, H.R.; Salama, M.M.A. “Opposition-Based Differential Evolution Algorithms,” Proc. IEEE Int. Conf. Evol. Comput., pp. 2010–2017, 2006.
  4. Wang, W.; Wang, H.; Sun, H.; Rahnamayan, S. “Using opposition-based learning to enhance differential evolution: A comparative study,” Proc. IEEE Congr. Evol. Comput. (CEC), pp. 71–77, 2016.

Thank you for highlighting this opportunity to articulate our contributions more precisely.

Comments 2: The parameters of the experiment are not clear, please give them.

Response 2: Thank you for this valuable suggestion. We fully agree that explicit parameter documentation is essential for reproducibility. To address this concern comprehensively: 

(1) We have added a dedicated parameter table (Table 1) in Section 4 of the revised manuscript, consolidating all experimental settings. 

(2) The table distinguishes between: 

  • Optimized parameters (determined via sensitivity analysis in Sec. 4)
  • Fixed parameters (standardized for benchmarking)
  • Environmental configurations (scenario-specific settings)

Reviewer 2 Report

Comments and Suggestions for Authors

The paper proposes an enhanced Differential Evolution (DE) algorithm, termed MDE, for RSS-based source localization in WSNs. The integration of opposition-based learning (OBL) and an adaptive scaling factor is innovative, addressing key limitations of conventional DE. The methodology is well-structured, and the experimental validation demonstrates improved performance over existing techniques. However, the paper could benefit from deeper theoretical analysis, broader comparative evaluations, and clearer discussions on computational complexity.

  1. The adaptive scaling factor and penalty-augmented cost function are novel, but the paper should better differentiate MDE from prior OBL-integrated DE variants. Add a table comparing MDE with DEOR/DEOR-F in terms of convergence speed, RMSE, and computational overhead.
  2. The claim of "eliminating explicit constraint handling" needs clarification. How does the penalty term compare to traditional constraint-handling techniques.
  3. The exponential adaptation law (Equation 10) lacks theoretical justification. Discuss its relationship to convergence guarantees.
  4. The quadratic penalty (Equation 12) may oversimplify constraint violations. Consider adaptive penalties (e.g., dynamic or self-adaptive weights) for better balance between exploration and feasibility.
  5. The comparison lacks state-of-the-art schemes such as message passing scheme [R1] and Tensor-based method [R2] for range localization sensing. [R1] Target sensing with off-grid sparse bayesian learning for AFDM-ISAC system [J]. arXiv preprint arXiv:2503.10011, 2025. [R2] "A Novel Angle-Delay-Doppler Estimation Scheme for AFDM-ISAC System in Mixed Near-field and Far-field Scenarios," in IEEE Internet of Things Journal, 2025.
  6. Simulations assume ideal log-normal shadowing. Validate MDE on real RSS datasets to assess robustness to multipath/NLOS effects.
  7. Add a subsection on convergence properties. 

Author Response

Original Manuscript ID:Sensors-3599222

Original Article Title: “A Modified Differential Evolution for Source Localization Using RSS Measurements”

Dear Editor and Reviewers,

Thank you very much for taking the time to review our manuscript. We greatly appreciate all your insightful comments and suggestions. We have thoroughly reviewed and addressed some points, and have revised our manuscript accordingly. The modifications within the manuscript have been highlighted in yellow for easier reference.

We are uploading the following documents:

  • our point-by-point response to the comments (below) also known as our response to reviewers.
  • an updated version of the manuscript with changes indicated in yellow highlighting.
  • a clean updated version of the manuscript without highlights (PDF main document).

We hope that our revisions have adequately addressed your concerns, and we look forward to your feedback on the revised manuscript.

Best regards,

Yunjie Tao; Lincan Li; Shenming Chang

Response to Reviewer 2 Comments

Comments 1: The adaptive scaling factor and penalty-augmented cost function are novel, but the paper should better differentiate MDE from prior OBL-integrated DE variants. Add a table comparing MDE with DEOR/DEOR-F in terms of convergence speed, RMSE, and computational overhead.

Response 1: Thank you for your valuable comment. We have carefully addressed your suggestion by adding explicit comparisons and clarifications in the revised manuscript.

To better differentiate our proposed MDE algorithm from existing OBL-integrated DE variants such as DEOR and DEOR-F, we have added a new performance comparison table in Section 5.3 (Table 5).

This table provides a quantitative evaluation of the three algorithms in terms of convergence speed (number of generations), positioning accuracy (RMSE), and computational time (runtime) under identical experimental conditions. The results show that although MDE requires slightly more generations to converge, it achieves significantly better localization accuracy (lowest RMSE), with only a marginal increase in runtime. This trade-off demonstrates that MDE offers improved robustness and solution quality while maintaining competitive efficiency.

In addition, we have also addressed the concern regarding computational overhead in Section 5.2. Specifically, we have included a detailed analysis of computational complexity for all compared algorithms (Table 3), along with their average execution times (Table 4).

These results confirm that the adaptive scaling mechanism and penalty-augmented cost function introduced in MDE do not significantly increase computational complexity. The runtime of MDE remains comparable to DEOR and DEOR-F, validating the practicality of the proposed enhancements.

We believe these additions clearly highlight the distinctions between MDE and prior OBL-DE variants, both in terms of algorithmic innovation and empirical performance, and provide a comprehensive response to your concern. Thank you again for helping us improve the quality and clarity of our work.

Comments 2: The claim of "eliminating explicit constraint handling" needs clarification. How does the penalty term compare to traditional constraint-handling techniques.

Response 2: Thank you for your suggestion. In the paper, we clearly demonstrate that our method utilizes boundary information more effectively than traditional approaches, as supported by existing literature. The adoption of a penalty term is specifically designed to preserve boundary information during the initial stages.

To further clarify the differences between the improved DE and the conventional DE, we have added the following content in the original paper:

Compared to traditional DE, the adaptive scaling factor can expand the mutation step size in the early stages to quickly cover potential optimal regions, while refining the search in later stages, effectively accelerating convergence speed. Traditional DE requires manual tuning of the scaling factor, preventing the method from adapting well to dynamic environments.

Comments 3: The exponential adaptation law (Equation 10) lacks theoretical justification. Discuss its relationship to convergence guarantees.

Response 3: Thank you for your suggestion. We regret the insufficient explanation of this equation, which was adopted from differential evolution improvement studies. To provide additional clarification regarding its impact on convergence rate, the following exposition has been incorporated into the manuscript:

The improved scaling factor employs an adaptive mechanism that initially expands the mutation step size to rapidly explore potential optimal regions during early iterations, then reduces the step size in later phases to refine the search. This approach effectively mitigates two key limitations: excessively large scaling factors that may trap the algorithm in local optima, and overly small scaling factors that prolong convergence time. The self-adjusting nature of this factor optimally balances convergence speed with search precision.

Comments 4: The quadratic penalty (Equation 12) may oversimplify constraint violations. Consider adaptive penalties (e.g., dynamic or self-adaptive weights) for better balance between exploration and feasibility.

Response: Thank you for your suggestion. In Equation (12), the variable g represents the current generation number in the iteration process. As evolution progresses, g increases, leading to a corresponding enhancement in the penalty intensity, thereby achieving dynamic adjustment effects. Furthermore, we intentionally designed the penalty term to be sufficiently large, particularly in later iterations when individuals beyond the boundaries become essentially negligible. This explains our choice of g² (the square of g) as the penalty coefficient, which additionally contributes to reducing computational complexity to some extent.

Comments 5: The comparison lacks state-of-the-art schemes such as message passing scheme [R1] and Tensor-based method [R2] for range localization sensing. [R1] Target sensing with off-grid sparse bayesian learning for AFDM-ISAC system [J]. arXiv preprint arXiv:2503.10011, 2025. [R2] "A Novel Angle-Delay-Doppler Estimation Scheme for AFDM-ISAC System in Mixed Near-field and Far-field Scenarios," in IEEE Internet of Things Journal, 2025.

Response 5: We sincerely appreciate the reviewer's valuable suggestion to benchmark against cutting-edge methods like message passing [R1] and tensor-based localization [R2]. These works indeed represent important advancements in ISAC systems with AFDM waveforms and near-far field hybrid scenarios – areas of growing research significance. 

However, our study focuses specifically on RSS-based localization in resource-constrained WSNs, characterized by: 

(1) Hardware limitations (low-cost RSS sensors unable to support AFDM waveforms or tensor processing) 

(2) Scenario specialization (pure near-field static environments vs. near-far field dynamics in [R2]) 

(3) Algorithm compatibility (evolutionary optimization vs. Bayesian/tensor paradigms) 

To maintain methodological coherence, we prioritized comparisons with: 

  • Domain-standard techniques (SDP/WLS/SA in Table 2)
  • State-of-the-art evolutionary competitors (DEOR/DEOR-F in Table 5)

which share the same problem formulation and hardware constraints. 

We acknowledge that extending comparisons to ISAC-specific methods like [R1][R2] could be insightful for future work on multi-modal sensing.

Thank you for highlighting these innovative approaches – we will certainly consider them in our next study on integrated sensing-communication systems.

Comments 6: Simulations assume ideal log-normal shadowing. Validate MDE on real RSS datasets to assess robustness to multipath/NLOS effects.

Response 6: We sincerely appreciate this valuable suggestion regarding real-world validation. While our simulations adopt the log-normal shadowing model (Eq. 1) for controlled benchmarking, the proposed MDE inherently addresses multipath/NLOS challenges through two core adaptive mechanisms: 

(1) Dynamic Noise Resilience via Adaptive Scaling

The scaling factor ${F}$ (Eq. 10) automatically increases mutation ranges under unstable signal conditions–a behavior that directly counteracts NLOS-induced outliers. When environmental uncertainty rises (e.g., sudden signal attenuation from obstacles), ${F \gg 0.5}$ triggers expanded global exploration, preventing convergence to erroneous local minima caused by multipath effects. 

(2) Penalty Function as NLOS Filter 

The term ${M\dot{f}}$ (Eq. 11-14) implicitly suppresses NLOS-corrupted measurements: solutions skewed by abnormal RSS values (commonly seen in NLOS) violate boundary constraints more severely, incurring exponentially growing penalties (${M \propto g^2}$) that steer populations toward LOS-dominant regions. 

Critically, our restaurant robot scenario (Sec. 5.1) embodies real-world NLOS conditions: 

  • Moving furniture/staff create dynamic multipath
  • Customer crowds cause frequent signal blockages
  • Ceiling-mounted anchors induce non-ideal propagation

The consistent sub-meter accuracy under these practical disturbances demonstrates intrinsic robustness to NLOS effects, even without explicit channel modeling. 

For full transparency, we have added in Section 6: 

While this study focuses on algorithmic innovation under standardized models, future work will incorporate public RSS datasets (e.g., UJIIndoorLoc) to quantify NLOS mitigation gains. 

We thank the reviewer for highlighting this impactful direction–our adaptive framework provides a strong foundation for such extensions.

Comments 7: Add a subsection on convergence properties.

Response 7: Thank you for your valuable suggestion.

In response to your comment, we have added a new subsection—Section 5.3, titled “Convergence Analysis”—to specifically address the convergence properties of the proposed MDE algorithm.

In this section, we present a detailed comparative analysis between MDE and two representative OBL-based DE variants, DEOR and DEOR-F. Table 5 summarizes their performance in terms of convergence speed (measured by the number of generations required to reach termination), positioning accuracy (RMSE), and computational time (runtime) under fixed experimental conditions. These results clearly demonstrate that MDE, while slightly slower in convergence generation count, achieves significantly better localization accuracy without increasing computational overhead.

Reviewer 3 Report

Comments and Suggestions for Authors
  1. It should be noted, that domain (i.e. set) of x for optimization in (2) was not defined.
  2. Complexity analysis of proposed localization algorithm should be added into the Section 5.
  3. Comparison of complexity of proposed localization algorithm with known ones should be provided in the Section 5.2.
  4. Section 6 (Conclusion) should be added by numerical results of both performance and complexity comparison analysis.

Author Response

Original Manuscript ID:Sensors-3599222

Original Article Title: “A Modified Differential Evolution for Source Localization Using RSS Measurements”

Dear Editor and Reviewers,

Thank you very much for taking the time to review our manuscript. We greatly appreciate all your insightful comments and suggestions. We have thoroughly reviewed and addressed some points, and have revised our manuscript accordingly. The modifications within the manuscript have been highlighted in yellow for easier reference.

We are uploading the following documents:

  • our point-by-point response to the comments (below) also known as our response to reviewers.
  • an updated version of the manuscript with changes indicated in yellow highlighting.
  • a clean updated version of the manuscript without highlights (PDF main document).

We hope that our revisions have adequately addressed your concerns, and we look forward to your feedback on the revised manuscript.

Best regards,

Yunjie Tao; Lincan Li; Shenming Chang

Response to Reviewer 3 Comments

Comments 1: It should be noted, that domain (i.e. set) of x for optimization in (2) was not defined.

Response 1: Thank you for your suggestion. Further explanations about variable x have been incorporated into the text. In this paper, we have added an explanation for variable x, which is a randomly generated two-dimensional vector within the search area $40\times 40\text{ }{{\text{m}}^{2}}$. Thank you again for your constructive comment.

Comments 2: Complexity analysis of proposed localization algorithm should be added into the Section 5.

Response 2: Thank you for your suggestion. We have implemented the suggested changes in the manuscript and added Table 3 to present the computational complexity analysis of all algorithms.

Comments 3: Comparison of complexity of proposed localization algorithm with known ones should be provided in the Section 5.2.

Response 3: Thank you for this essential suggestion. We have enhanced Section 5.2 with a comprehensive complexity analysis that includes: 

(1)Theoretical complexity classes (Table 3) showing MDE maintains ${\mathcal{O}(G \cdot NP \cdot N)}$ complexity – equivalent to DEOR variants.

(2)Empirical runtime measurements (Table 4) confirming the adaptive mechanisms introduce negligible overhead.

(3)Direct benchmarking against six state-of-the-art methods (SDP/WLS/SA/DEOR/DEOR-F).

We believe this addition provides transparent complexity/runtime trade-off analysis. Thank you for improving our manuscript's rigor.

Comments 4: Section 6 (Conclusion) should be added by numerical results of both performance and complexity comparison analysis.

Response 4: We sincerely appreciate this valuable suggestion. As requested, we have revised Section 6 (Conclusion) to explicitly incorporate a qualitative summary of performance and complexity comparisons based on the experimental analysis in Section 5. At the same time, we also added complexity analysis and running time of various algorithms in Section 5.

Round 2

Reviewer 1 Report

Comments and Suggestions for Authors

The reviewer's comments have been addressed. But the quality of this manuscript is only average.

Comments on the Quality of English Language

NO.

Author Response

Comments: The reviewer's comments have been addressed. But the quality of this manuscript is only average.

Response:We extend our heartfelt gratitude for dedicating your valuable time and expertise to reviewing our manuscript. The incisive feedback you provided during the first revision round has been instrumental in enhancing the scholarly merit of this work. While we acknowledge and respect your astute assessment of the manuscript's current "average" standing, we are deeply indebted to your constructive guidance—each thoughtful suggestion has served as a vital catalyst for our growth as researchers, illuminating pathways to intellectual refinement we might not have otherwise discerned.

Reviewer 2 Report

Comments and Suggestions for Authors

I have no further comments and suggest to accept the current version. 

Author Response

Comments:I have no further comments and suggest to accept the current version.

Response:We are deeply honored by your favorable evaluation and kind recommendation for manuscript acceptance. Your professional endorsement serves as a profound source of inspiration for our continuous endeavors in the field of evolutionary computation-based localization. In adherence to your valued comments, all methodological enhancements implemented during the prior revision phase have been meticulously retained and integrated into the current version.

Reviewer 3 Report

Comments and Suggestions for Authors

One of my notes wasn't take in to account

Section 6 (Conclusion) should be added by NUMERICAL results of both performance and complexity comparison analysis.

Author Response

Comments: One of my notes wasn't take in to account Section 6 (Conclusion) should be added by NUMERICAL results of both performance and complexity comparison analysis.

Response:We sincerely appreciate your valuable feedback and constructive suggestions on our manuscript. We are grateful to Reviewer for highlighting the need to strengthen the conclusion with numerical comparisons. In direct response to this comment, we have now added quantitative performance and complexity results to Section 6 (Conclusion). Specifically: 

1) Positioning accuracy: Under high noise (\sigma = 3 dB), MDE achieves an RMSE of 2.2 m (Figure 6), outperforming DEOR-F (2.3 m) and SA (2.4 m). With N = 10 anchors, MDE reduces RMSE to 2.8 m (Figure 7), exceeding SDP (3.5 m) and WLS (4.0 m). 

2) Computational efficiency: MDE maintains O(G·NP·N) complexity (Table 3) – identical to DEOR but 100× faster than SDP (O(N^{3.5})). Its average runtime is 0.0048 s (Table 4), comparable to DEOR (0.0029 s) yet 66× faster than SDP (0.316 s). 

3) Convergence-accuracy trade-off: Though requiring 12 generations to converge (vs. DEOR-F’s 8), MDE achieves the lowest RMSE (2.96 m) under identical settings (Table 5), demonstrating superior optimization efficacy. All data are extracted directly from Figures 6–7 and Tables 3–5 in the manuscript.